# Decoupling Task and Behavior: A Two-Stage Reward Curriculum in Reinforcement Learning for Robotics

## Abstract

Deep Reinforcement Learning is a promising tool for robotic control, yet practical application is often hindered by the difficulty of designing effective reward functions. Real-world tasks typically require optimizing multiple objectives simultaneously, necessitating precise tuning of their weights to learn a policy with the desired characteristics. To address this, we propose a two-stage reward curriculum where we decouple task-specific objectives from behavioral terms. In our method, we first train the agent on a simplified task-only reward function to ensure effective exploration before introducing the full reward that includes auxiliary behavior-related terms such as energy efficiency. Further, we analyze various transition strategies and demonstrate that reusing samples between phases is critical for training stability. We validate our approach on the DeepMind Control Suite, ManiSkill3, and a mobile robot environment, modified to include auxiliary behavioral objectives. Our method proves to be simple yet effective, substantially outperforming baselines trained directly on the full reward while exhibiting higher robustness to specific reward weightings.

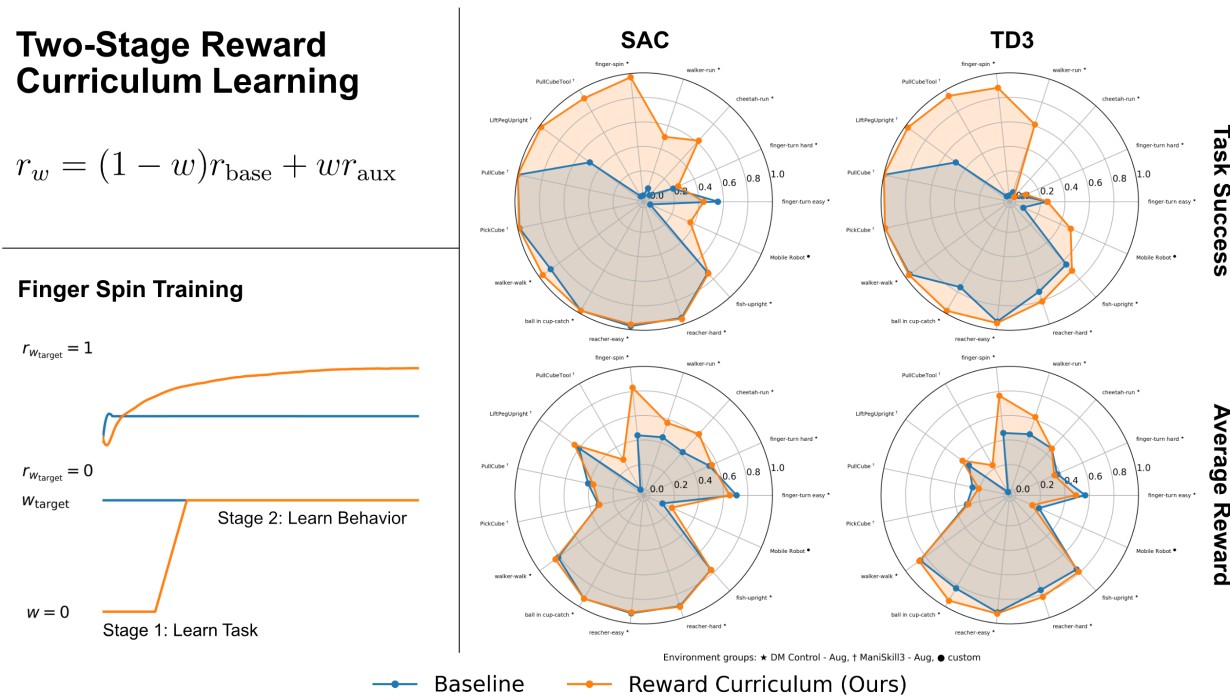

Figure 1: Comparison of baseline TD3 and SAC algorithms against reward curriculum versions (RC-TD3 and RC-SAC). Results are averaged over the last 50k training steps and 3 random seed using target weight $w_{\text{target}} = 0.5$ for DM Control, $w_{\text{target}} = 0.25$ for ManiSkill3 and $w_{\text{target}} = 0.75$ for Mobile Robot.

# 1 Introduction

Reinforcement Learning (RL) has emerged as a powerful paradigm in the field of robotic control, offering the promise of adaptable and efficient solutions to complex problems. RL has demonstrated its potential to learn optimal policies in a wide range of applications, such as manipulation (Gu et al., 2016; Kalashnikov et al., 2018; Leyendecker et al., 2022; Han et al., 2023) or mobile robotics (Lillicrap, 2015; Zhu et al., 2017; Rodriguez-Ramos et al., 2018; Zhu & Zhang, 2021; Sang & Wang, 2022). However, as we transition from carefully curated benchmarks to realistic scenarios, a significant gap emerges, highlighting the challenges of applying RL in practical settings.

One of the primary challenges in realistic applications lies in the complexity of the environment and the multiplicity of objectives. While classical RL problems often focus on a single, well-defined goal, real-world scenarios typically involve multiple, sometimes conflicting rewards. For instance, a mobile robot might need to navigate to a goal location while simultaneously avoiding obstacles, maintaining a specific velocity, and ensuring a smooth trajectory (Zhang et al., 2023; Ceder et al., 2024). This multi-objective nature of real-world problems poses a significant challenge to traditional RL approaches.

Formulating an effective reward function for robotic tasks is non-trivial and can often result in undesired behaviors (Booth et al., 2023; Knox et al., 2023; Knox & MacGlashan, 2024). Moreover, optimizing multiple weighted rewards can be challenging due to the presence of local optima, where policies might satisfy only a subset of objectives (e.g., minimizing energy consumption by remaining stationary) without learning the intended task, which is referred to as reward hacking. We call such potentially conflicting objectives complex reward functions, where the complexity lies in the presence of strong local optima for undesired behaviors. Notably, our setting focuses on combining multiple objectives into a single reward function to learn one policy, distinct from multi-objective reinforcement learning (MORL), which typically considers a set of Pareto-optimal solutions for multiple objectives (Hayes et al., 2022).

A line of work that has emerged to tackle challenging RL problems is curriculum learning (Bengio et al., 2009; Narvekar et al., 2020). Inspired by how animals can be trained to learn new skills (Skinner, 1958), learning proceeds from simple problems to gradually more challenging ones. In the realm of RL, such curricula are often hand-designed and have been successfully employed in several applications in robotic control (Sanger, 1994; Hwangbo et al., 2019; Leyendecker et al., 2022). More recently, several methods for automatic curriculum design have emerged that are able to learn curriculum policies (Narvekar & Stone, 2019) or that break down problems into smaller ones (Andrychowicz et al., 2017; Fang et al., 2019) for more effective learning. Further, automatic curricula are used in sparse or no reward settings as intrinsically motivated exploration that encourages reaching diverse states (Bellemare et al., 2016; Pathak et al., 2017; Burda et al., 2018; Shyam et al., 2019). Nevertheless, such methods have been explored less for reward functions and often focus on task difficulty or goal location.

To address the challenge of learning with complex reward functions, we propose leveraging curriculum learning. We introduce a novel two-stage reward curriculum that distinguishes between task and behavior-related reward terms. In the first phase, only a subset of task-related rewards is used for training to simplify the discovery of successful trajectories. When the policy has converged sufficiently, the second phase is initiated by transitioning towards optimizing the full reward, including behavior-related rewards. Additionally, our method allows for sample-efficient reuse of collected trajectories by incorporating two rewards in the replay buffer such that samples from the first phase can be reused for training with an updated reward in the second phase.

To analyze the efficacy of our method, we test it on several robotics benchmarks where we augment the reward function to include behavioral terms that are often desired for robotics (for instance minimizing jerk), such as the DeepMind (DM) Control Suite (Tassa et al., 2018), ManiSkill3 (Tao et al., 2025), and a mobile robot environment (Ceder et al., 2024). Overall, it substantially outperforms a baseline trained on the full reward from the beginning (see Fig. 1) and shows a greater robustness to varying weights of auxiliary behavioral terms, thus simplifying design choices for the experimenter. Notably, it works best in cases where the task is learnable, but the auxiliary terms are in conflict with the exploration needed to master it.

Our contributions can be summarized as follows:

1. We introduce a novel two-stage reward curriculum to effectively learn complex rewards by first learning the task and then adding behavioral rewards. The curriculum is integrated into two RL methods, one based on SAC and the other based on TD3.

2. In ablation studies, we compare different strategies of when to switch between curriculum phases, how to transition towards optimizing the full reward, and the importance of reusing samples between phases.

3. We extensively evaluate our method on several realistic robotics environments where it consistently outperforms a baseline trained on the full reward, and we show its robustness to different reward weights, alleviating the need for precise tuning.

## 2 Related Work

A parallel line of work that has been studied extensively addresses auxiliary terms via Safe Reinforcement Learning (Berkenkamp et al., 2017; Turchetta et al., 2020; Varga et al., 2021; As et al., 2024; Gu et al., 2024). These methods typically formulate auxiliary objectives as hard constraints (e.g., safety budgets) that must be satisfied below a specific threshold. However, many behavioral objectives, such as energy efficiency or motion smoothness, do not have natural binary thresholds. Rather, they are soft objectives that should be minimized continuously. Unlike constrained approaches which target a feasible set, our method treats these terms as a continuous trade-off against task performance, allowing the agent to find the optimal balance without requiring pre-defined constraint thresholds or complex Lagrangian optimization.

Although less prevalent, several works have employed reward curricula in reinforcement learning (RL) using subsets of the true reward. For instance, Hwangbo et al. (2019) utilize an exponential reward curriculum, initially focusing on locomotion and gradually increasing the weight of other cost terms to refine the behavior. Similarly, Leyendecker et al. (2022) observe that RL agents optimized on complex reward functions with multiple constraints are highly sensitive to individual reward weights, leading to local optima when directly optimizing the full reward. They successfully learn a policy by gradually increasing constraint weights based on task success. Furthermore, Pathare et al. (2025) employ a three-stage reward curriculum, incrementally increasing complexity and realism by incorporating additional terms. However, they find reward curriculum learning largely ineffective. To the best of our knowledge, we are the first to systematically investigate a two-stage reward curriculum.

## 3 Problem Formulation

We formulate the problem as a Markov Decision Process (MDP) defined by the tuple $\langle \mathcal{S}, \mathcal{A}, \mathcal{P}, r, p_0, \gamma \rangle$. $\mathcal{A}$ represents the action space, $\mathcal{S}$ the state space, $\mathcal{P} : \mathcal{S} \times \mathcal{A} \times \mathcal{S} \to [0,1]$ is the state transition function, $p_0 : \mathcal{S} \to [0,1]$ the initial state distribution and $\gamma \in [0,1)$ a discount factor. The reward is denoted by $r$; we omit its explicit dependence on state and action $r(s,a)$ for a less clustered notation.

The objective of RL is to maximize the expected return, $\mathbb{E}[G_n]$, where $G_n = \sum_{k=n}^{N} \gamma^{k-n} r_k$ is the cumulative discounted reward, with $n$ being the current step and $N$ the maximum steps per episode. A key concept for achieving this goal is the action-value function, commonly referred to as the $Q$-function, which represents the expected return when taking action $a$ in state $s$. The Q-function is parameterized by weights $\phi$, and denoted as $Q_\phi(s,a)$.

In our case, we consider any control problem with several reward terms, where each can be categorized either as a base reward $r_{\text{base}}$ if it helps learning the task (for instance reward shaping terms), and auxiliary rewards $r_{\text{aux}}$, which specify the desired behavior. The reward is then given by

$$r_w = (1 - w) \cdot r_{\text{base}} + w \cdot r_{\text{aux}} \tag{1}$$

with $w \in [0,1]$. Thus, increasing $w$ keeps the overall reward magnitude stable but will shift the focus from $r_{\text{base}}$ to $r_{\text{aux}}$. Further, we assume there is a target reward function defined by $w_{\text{target}} \in [0,1)$ which remains

smaller than 1, such that $r_{\text{base}}$ is never fully neglected. While we require having access to individual reward terms, the framework works for any number of terms. They simply need to be assigned to either the base or auxiliary reward.

The goal is to learn a policy $\pi_\theta(a|s)$ parametrized by $\theta$ that learns to complete the primary task (encoded by $r_{\text{base}}$) while optimizing the auxiliary objectives $r_{\text{aux}}$. A key requirement is robustness to the target weight $w_{\text{target}}$, eliminating the need for precise tuning. This challenge is common in robotics, for instance when training an energy-efficient policy for a robotics manipulator. If the weight on the efficiency term is too high, the penalties effectively discourage exploration, preventing the agent from ever learning the manipulation task. Conversely, if the weight is too low, the auxiliary objective is ignored. Our method aims to resolve this conflict by decoupling task acquisition from auxiliary optimization, avoiding the local optima often caused by conflicting reward signals.

## 4 Reward Curriculum Framework

We propose a novel two-stage reward curriculum to effectively learn complex reward functions in a sample-efficient manner. While this framework is compatible with any off-policy RL algorithm, we demonstrate its effectiveness on **RC-SAC** (based on Soft-Actor Critic (Haarnoja et al., 2018)) and **RC-TD3** (based on Twin-Delayed DDPG (Fujimoto et al., 2018)).

The core principle is to dynamically adjust the weighting $w$ in Eq. 1 to shift focus from fundamental task learning to the combined objective. In the first phase ($k = 0$), the agent trains exclusively on the base reward ($w = 0$) to acquire essential task behaviors without interference from auxiliary terms. The second phase ($k = 1$) is initiated automatically when a transition criterion triggers (see Section 4.4).

Upon switching to the second phase, we introduce the auxiliary reward by annealing $w$ from 0 towards the target weight $w_{\text{target}}$. The curriculum weight $w_t$ at any global step $t$ is defined as:

$$w_t = \begin{cases} 0 & \text{if } k = 0, \\ \alpha(\Delta t) \cdot w_{\text{target}} & \text{if } k = 1, \end{cases} \qquad (2)$$

where $\Delta t = t - t_{\text{switch}}$ is the number of steps elapsed since the phase transition. The annealing factor $\alpha(\Delta t) \in [0, 1]$ increases from 0 to 1 over a fixed duration $T_{\text{ann}}$ (e.g., via a linear or cosine schedule) and remains at 1 thereafter (see Section 4.5). This smooth transition prevents sudden shocks to the value function estimates, allowing the policy to adapt to the auxiliary objectives while maintaining task performance.

---

**Algorithm 1** Two-Stage Reward Curriculum

1: **Input:** Policy $\pi_\theta$, Critic $Q_\phi$, Buffer $\mathcal{D}$
2: **Input:** Anneal duration $T_{\text{ann}}$, Target weight $w_{\text{target}}$
3: **Init:** Global step $t \leftarrow 0$, Switch step $t_{\text{switch}} \leftarrow \infty$
4: **Init:** Phase $k \leftarrow 0$, Metric history $\mathcal{M} \leftarrow \emptyset$
5: **while** $t < T_{\text{max}}$ **do**
6:     **for** env steps **do**
7:         Sample action $a_t \sim \pi_\theta(\cdot|s_t)$
8:         Step env: $s_{t+1}, r_{\text{base}}, r_{\text{aux}}$
9:         $\mathcal{D} \leftarrow \mathcal{D} \cup \{(s_t, a_t, r_{\text{base}}, r_{\text{aux}}, s_{t+1})\}$
10:         $t \leftarrow t + 1$
11:     **end for**
12:     /* **Metric Tracking** */
13:     Update $\mathcal{M}$ with current $r_{\text{base}}$ and loss metrics
14:     **if** $k = 0$ **and** `CheckCriterion`$(\mathcal{M})$ **then**
15:         $k \leftarrow 1$;    $t_{\text{switch}} \leftarrow t$
16:     **end if**
17:     $w \leftarrow 0$
18:     **if** $k = 1$ **then**
19:         $\Delta t \leftarrow t - t_{\text{switch}}$
20:         $\alpha \leftarrow$ `AnnealFactor`$(\Delta t, T_{\text{ann}})$    ▷ $0 \rightarrow 1$
21:         $w \leftarrow \alpha \cdot w_{\text{target}}$
22:     **end if**
23:     **for** $G$ gradient steps **do**
24:         Sample batch $B \sim \mathcal{D}$
25:         /* **Reward Construction** */
26:         $r_w \leftarrow (1 - w) \cdot r_{\text{base}} + w \cdot r_{\text{aux}}$
27:         $B \leftarrow \{(s, a, r_w, s')\}$
28:         Update $\phi \leftarrow \phi - \alpha_Q \nabla_\phi \mathcal{L}_Q(\phi, B)$
29:         Update $\theta \leftarrow \theta - \alpha_\pi \nabla_\theta \mathcal{L}_\pi(\theta, B)$
30:         Update target parameters
31:     **end for**
32: **end while**

---

The complete procedure is summarized in Algorithm 1. The function `CheckTransition` monitors performance metrics $\mathcal{M}$ (such as returns per reward term or actor losses) to trigger the phase switch. Parameters denoted with a bar (e.g., $\bar{\theta}$) indicate target networks updated via Polyak averaging.

### 4.1 Reward Curriculum RL Methods

In the following, we show how the reward curriculum integrates with the off-policy RL algorithms TD3 and SAC. Importantly, our framework is compatible with any off-policy algorithm and we pick TD3 and SAC as examples given their continued popularity.

**RC-TD3** Twin-Delayed DDPG (TD3) (Fujimoto et al., 2018) extends the Deep Deterministic Policy Gradient (DDPG) (Lillicrap, 2015) algorithm to enhance its stability by making use of two Q-functions, delay the policy updates, and add noise to target actions to avoid exploitation of Q-function errors.

The algorithm uses a deterministic policy, though Gaussian noise is added during data collection for exploration

$$a_t = \pi_\theta(s_t) + \epsilon \quad \epsilon \sim \mathcal{N}(0, \sigma)$$

A similar procedure using clipped noise is applied to the target action $\tilde{a}'$ to smooth the target value estimates

$$\tilde{a}' = \pi_{\bar{\theta}}(s') + \epsilon \quad \epsilon \sim \text{clip}(\mathcal{N}(0, \tilde{\sigma}), -c, c)$$

where $\pi_{\bar{\theta}}$ is a target policy. The curriculum-aware Q-targets are computed as

$$y(r_w, s') = r_w + \gamma \min_{i=1,2} Q_{\bar{\phi}_i}(s', \tilde{a}')$$

The critic networks are updated by minimizing the Smooth L1 loss (Girshick, 2015) with respect to these targets in order to stabilize training

$$\mathcal{L}_Q(\phi_i) = \mathbb{E}_{s,a,r_w,s' \sim \mathcal{B}} \left[ \text{SmoothL1}(Q_{\phi_i}(s,a) - y(r_w, s')) \right]$$

The policy is then updated by minimizing the negative Q-value estimated by the first critic

$$\mathcal{L}_\pi(\theta) = \mathbb{E}_{s \sim \mathcal{B}} \left[ -Q_{\phi_1}(s, \pi_\theta(s)) \right] \tag{3}$$

Importantly, the policy is not updated in every iteration. Instead it is delayed and is only updated in every second iteration. In all experiments we set $\tilde{\sigma} = 0.2$, $c = 0.5$ while we use $\sigma = 0.1$ for DM Control and ManiSkill3, and in MobileRobot we anneal $\sigma$ from 9.0 to 1.0 over the first 250k steps for enhanced exploration.

**RC-SAC** Instead of solely optimizing the expected return, SAC makes use of the maximum entropy objective (Ziebart, 2010) such that a policy additionally tries to maximize its entropy $\mathcal{H}$ at each state which is defined as

$$\pi_\theta^* = \arg\max_{\pi_\theta} \sum_t \mathbb{E}_{(s_t, a_t) \sim \rho_{\pi_\theta}} \left[ r_w(s_t, a_t) + \alpha \mathcal{H}(\pi_\theta(\cdot | s_t) \right]$$

where $\alpha$ is a parameter to control the policy temperature, i.e. the relative importance of the entropy term. SAC makes use of two Q-functions parameterized by $\phi_1$ and $\phi_2$ which are updated using the Smooth L1 loss (Girshick, 2015) as

$$\mathcal{L}_Q(\phi_i) = \mathbb{E}_{s,a,r_w,s' \sim \mathcal{B}} \left[ \text{SmoothL1}(Q_{\phi_i}(s,a) - y(r_w, s')) \right]$$

where $i$ is the Q-function index and the targets are computed as

$$y(r_w, s') = r_w + \gamma \left[ \min_{i=1,2} Q_{\bar{\phi}_i}(s', \tilde{a}') - \alpha \log \pi_\theta(\tilde{a}' | s') \right]$$

Importantly, $\tilde{a}' \sim \pi_\theta(\cdot | s')$ is newly sampled during training and $Q_{\bar{\phi}_i}$ is the $i$th target Q-function. Note that instead of optimizing for a general reward $r$, we use $r_w$ in this step which changes depending on the phase. The policy update is given by

$$\mathcal{L}_\pi(\theta) = \mathbb{E}_{s \sim \mathcal{B}} \left[ -\min_{i=1,2} Q_{\phi_i}(s, \tilde{a}_\theta(s)) - \alpha \log \pi_\theta(\tilde{a}_\theta(s) | s) \right] \tag{4}$$

where $\tilde{a}_\theta(s)$ is sampled from $\pi_\theta(\cdot|s)$ via the reparameterization trick (Ruiz et al., 2016). We make use of the entropy-constraint variant of SAC as described in (Haarnoja et al., 2018), where $\alpha$ is updated as

$$\mathcal{L}(\alpha) = \mathbb{E}_{a \sim \pi_\theta} \left[ -\alpha \log \pi_\theta(a|s) - \alpha \tilde{\mathcal{H}} \right]$$

where $\tilde{\mathcal{H}}$ is the target entropy set to $-\dim(\mathcal{A})$. An important parameter in SAC is the initial value for $\alpha$, which we denote as $\alpha_{\text{init}}$. It determines the importance of the entropy term and, thus how much the agent explores different states. When $\alpha$ converges to low values, the agent focuses mainly on the objective instead and becomes more deterministic. For our experiments, we set $\alpha_{\text{init}} = 1.0$.

## 4.2 Learning Intuition

To understand how the reward curriculum simplifies learning, we can model the learning process as overcoming exploration barriers through a sequential restriction of the objective. First, let $J_{\text{base}}(\theta) = \mathbb{E}_{s \sim \mathcal{D}} \left[ Q_{\text{base}}(s, \pi_\theta(s)) \right]$ be the task objective and $J_{\text{aux}}(\theta) = \mathbb{E}_{s \sim \mathcal{D}} \left[ Q_{\text{aux}}(s, \pi_\theta(s)) \right]$ the auxiliary objective, then for a curriculum step $t$ the combined objective is parameterized by $w_t \in [0, w_{\text{target}}]$ as

$$J_{w_t}(\theta) = (1 - w_t)J_{\text{base}}(\theta) + w_t J_{\text{aux}}(\theta) \tag{5}$$

If optimized directly with $w_{\text{target}}$ learning conflicts can arise when auxiliary tasks penalize exploration which can create a strong local minima around trivial behaviors (such as standing still for effort penalties).

Instead, in our proposed reward curriculum the optimizer strictly maximizes $J_{\text{base}}(\theta)$ during phase 1. Thus the agent can first freely explore the state-action space to find task-solving behaviors without behavioral penalties. This is done until the policy converges towards a "Task Manifold" ($\mathcal{M}_{\text{task}}$) which we define as the region in parameter space where the task is solved and $\nabla_\theta J_{\text{base}}(\theta)$ becomes stable.

When $\mathcal{M}_{\text{task}}$ is established, phase 2 is initialize by annealing $w$ towards $w_{\text{target}}$. Intuitively, by slowly increase the weight of $J_{\text{aux}}(\theta)$ the policy is pushed along $\mathcal{M}_{\text{task}}$ towards a sub-region that simultaneously maximizes the behavioral terms. Thus the curriculum smoothly encourages solutions that are close to $\mathcal{M}_{\text{task}}$, bypassing trivial local optima that trap baselines trained directly on $J_{w_{\text{target}}}(\theta)$.

## 4.3 Target Stability

In order for the actor to track the moving objective in Eq. 5, the critic's Temporal Difference (TD) targets must remain stable. By assuming that we take small weight increments $\Delta w$ per step, we can bound the maximum shift in the immediate reward by the maximum reward divergence $R_{\text{max}} = \max_{s,a} |r_{\text{aux}} - r_{\text{base}}|$ (assuming that $r_{\text{aux}}$ is bounded) as

$$|\Delta r| \leq \Delta w \cdot R_{\text{max}} \tag{6}$$

Which consequently implies that the Bellman target shift $\Delta y$ over an infinite horizon is bounded by the geometric series

$$|\Delta y| \leq \sum_{k=0}^{\infty} \gamma^k |\Delta r| \leq \frac{\Delta w \cdot R_{\text{max}}}{1 - \gamma} \tag{7}$$

We can see that if $\Delta w$ is small then also $\Delta y$ is small, implying that the Q-learning targets are Lipschitz continuous with respect to $w$, thus satisfying the stability conditions required for the actor to track the manifold. However, when $\gamma \to 1$, the bound becomes quite loose, for instance for a common choice such as $\gamma = 0.99$ it is multiplied by 100. Thus an instantaneous switch from $w = 0$ to $w_{\text{target}}$ could shatter the Bellman target and annealing is necessary to prevent this. Further, we implicitly assumes that the actor is smooth with respect to the change in critic's targets which might not always be true in practice. Nevertheless, a curriculum that anneals $\Delta w$ sufficiently slow should compensate for that.

## 4.4 Phase Switch Mechanisms

A critical component of the reward curriculum is determining the optimal timestep $t_{\text{switch}}$ to transition from the initial phase ($k = 0$) to the combined reward phase ($k = 1$). We evaluate several strategies for the function `CheckCriterion`($\mathcal{M}$) (line 12 in Alg. 1):

**Actor fit threshold:** One hypothesis is that the actor must sufficiently learn the base objective before tackling auxiliary tasks. We quantify this by monitoring the actor's ability to minimize the critic's value estimate. For TD3, this metric is the standard actor loss in Eq. 3 and for SAC we utilize the policy loss in Eq. 4 without the entropy term to isolate the exploitation capability. We define a "good fit" threshold $\Gamma_{\text{fit}}$ that triggers if the actor loss $\mathcal{L}_\pi$ remains below this threshold for a window of $m$ consecutive steps:

$$\texttt{CheckCriterion}(\mathcal{M}) \triangleq \Big(\forall i \in [t-m,t] : \mathcal{L}_\pi(i) < \Gamma_{\text{fit}}\Big)$$

This ensures the policy has stably converged to a local optimum with respect to the base reward critic. In experiments including this switch we set $\Gamma_{\text{fit}} = -50$ and $m = 20$.

**Base reward threshold:** Alternatively, we can use the extrinsic performance directly as a proxy for learning progress. This strategy triggers the phase switch once the agent achieves a target reward $\Gamma_{r_{\text{base}}}$ on the base task. Let $\bar{r}_{\text{base},t}$ be the average of the the base reward at step $t$ for the last 1000 steps. The condition is formally defined as:

$$\texttt{CheckCriterion}(\mathcal{M}) \triangleq \Big(\bar{r}_{\text{base},t} \geq \Gamma_{r_{\text{base}}}\Big)$$

For the experiments including this switch method we set $\Gamma_{r_{\text{base}}} = 0.5$ (with 1.0 being the highest possible base reward for DM control environments).

**Base reward convergence:** While the two options above are intuitive, their hyperparameters $\Gamma_{\text{fit}}$ and $\Gamma_{r_{\text{base}}}$ vary across environments and algorithms, thus often require manual tuning. To address this we propose another mechanism that is based on convergence of $r_{\text{base}}$. The intuition is to initiate the second phase only once the agent's performance on the base task has plateaued, regardless of the absolute return value. We again use the average base reward $\bar{r}_{\text{base},t}$ over the last 1000 steps followed by a 20-point rolling average to further smooth the curve to obtain $\tilde{r}_{\text{base},t}$. We estimate the slope $\beta_t$ of the performance trend by fitting a Huber Regressor (Owen, 2007) to $\tilde{r}_{\text{base},t}$ over the recent history $h$. Unlike Ordinary Least Squares (OLS), which penalizes outliers quadratically, the Huber Regressor minimizes a hybrid loss function: it applies a squared loss to small errors (inliers) and a linear loss to errors exceeding a threshold $\epsilon_{\text{Huber}}$ (outliers) which we set to the default of 1.35. This linear tail ensures that the high variance and sporadic reward spikes inherent in stochastic RL training do not disproportionately skew the estimated trend. The switch is triggered when the slope indicates a plateau (i.e., falls below a small value $\epsilon_{\text{slope}}$) for $m$ steps, provided the agent has improved upon the random policy baseline $\bar{R}_{\text{init}}$:

$$\texttt{CheckCriterion}(\mathcal{M}) \triangleq \Big((\beta_t < \epsilon_{\text{slope}})$$
$$\wedge (\bar{r}_{\text{base},t} > 1.5 \cdot \bar{r}_{\text{base,init}})\Big)$$

The second condition $(\bar{r}_{\text{base},t} > 1.5 \cdot \bar{r}_{\text{base,init}})$, where $\bar{r}_{\text{base,init}}$ is collected with a random policy for the first 10k steps, prevents premature transitions during the initial exploration phase where the slope might be flat simply because learning has not yet begun. Further, we prevent any transition before recent history is filled.

We found that the method is robust to the exact hyperparameters chosen (see Appendix B.1), and we set $\epsilon_{\text{slope}} = 0.001$, $h = 75$, and $m = 20$ for all of our experiments.

## 4.5 Transition Dynamics and Annealing

Upon triggering the second phase of the curriculum, the objective function transitions from the base reward to the full task reward. We investigate the impact of the transition dynamics on agent stability by comparing three annealing schedules for the weight vector $w$: instantaneous switching (step function), linear interpolation, and cosine annealing. These schedules dictate the `AnnealFactor` in line 18 of Alg 1.

We further ablate the duration of this transition (measured in environment steps). We hypothesize a trade-off: abrupt changes in the reward function (instant switching) can induce large shifts in the Q-values, potentially

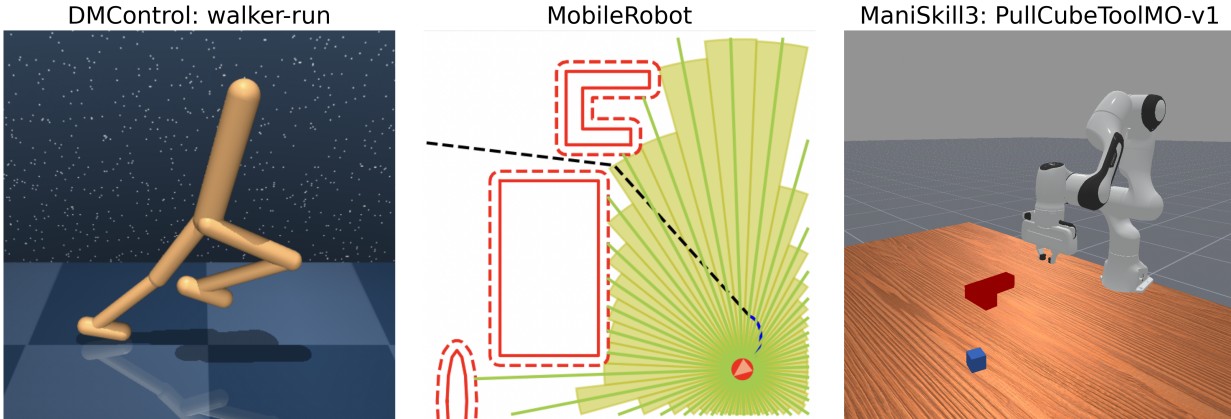

Figure 2: Exemplary environments from DM control (walker-run), MobileRobot, and ManiSkill3 (PullCubeTool-v1).

destabilizing the policy. Conversely, overly extended annealing may waste computational resources. However, given that the base reward is a subset of the full reward ($r_{\text{base}} \subset r_{w_{\text{target}}}$), we posit that the gradients are partially aligned, potentially allowing for more sudden transition schedules than would be required for disjoint tasks.

### 4.6 Reuse of past experience

Another key aspect of our framework is leveraging past experiences after a phase switch. Specifically, we store tuples of the form $\{(s_t, a_t, r_{\text{base}}, r_{\text{aux}}, s_{t+1})\}$ in the replay buffer. When transitioning to the second phase, we reuse experiences from the first phase by calculating the reward $r_w$ with the current $w$ for gradient updates. This mechanism aims to stabilize training by populating the replay buffer with a diverse set of samples, thereby facilitating sample-efficient learning. Notably, this approach is only compatible with off-policy RL algorithms.

## 5  Experiments

To evaluate the efficacy of our method we test RC-SAC and RC-TD3 on i) 12 DM control suite environments (Tassa et al., 2018) augmented to include an acceleration penalty to encourage smoothness, ii) a MobileRobot environment (Ceder et al., 2024) with several task-related and behavior-related reward terms, and iii) 4 robot manipulation environments from ManiSkill3 (PickCube-v1, LiftPegUpright-v1, PullCubeTool-v1, PullCube-v1) (Tao et al., 2025) with additional behavior-related rewards to reduce jerk, effort and action smoothness (see Fig. 2 for some exemplary environments). Introducing those terms makes the benchmarks relevant for real-life robotics, as, for instance, reducing jerk is vital when deploying a policy to hardware. Details about the environments are described in Appendix A. Note that we exclude the sparse outcome reward for MobileRobot from the curriculum (see Appendix A.4 for details).

As neural network architecture, we use two fully connected layers, each with 256 hidden units and ReLU activation. We train for 2000 [k] environments steps in the DM control and ManiSkill3 environments and for 1000 [k] for MobileRobot. In DM control and mobile robot environments we use a replay ratio of 1, where we first sample 1000 environment steps and then train for the same number of gradient steps while we use a ratio of 0.5 for ManiSkill3. Our replay buffer has a capacity of $1_000\,000$ samples and we set all learning rates ($\lambda_Q$, $\lambda_\pi$, $\lambda_\alpha$) to $3.0 \times 10^{-4}$. For DM Control and MobileRobot we set $\tau_{\text{targ}} = 0.995$ with a batch size of 128 and for ManiSkill3 $\tau_{\text{targ}} = 0.99$ with a batch size of 1024. Further, we set $\gamma = 0.99$ and the maximum episodes steps to their defaults at 1000 for DM control, 300 for MobileRobot, and 50 for ManiSkill3. All results are computed for 3 random seeds on Nvidia T4 and A40 GPUs.

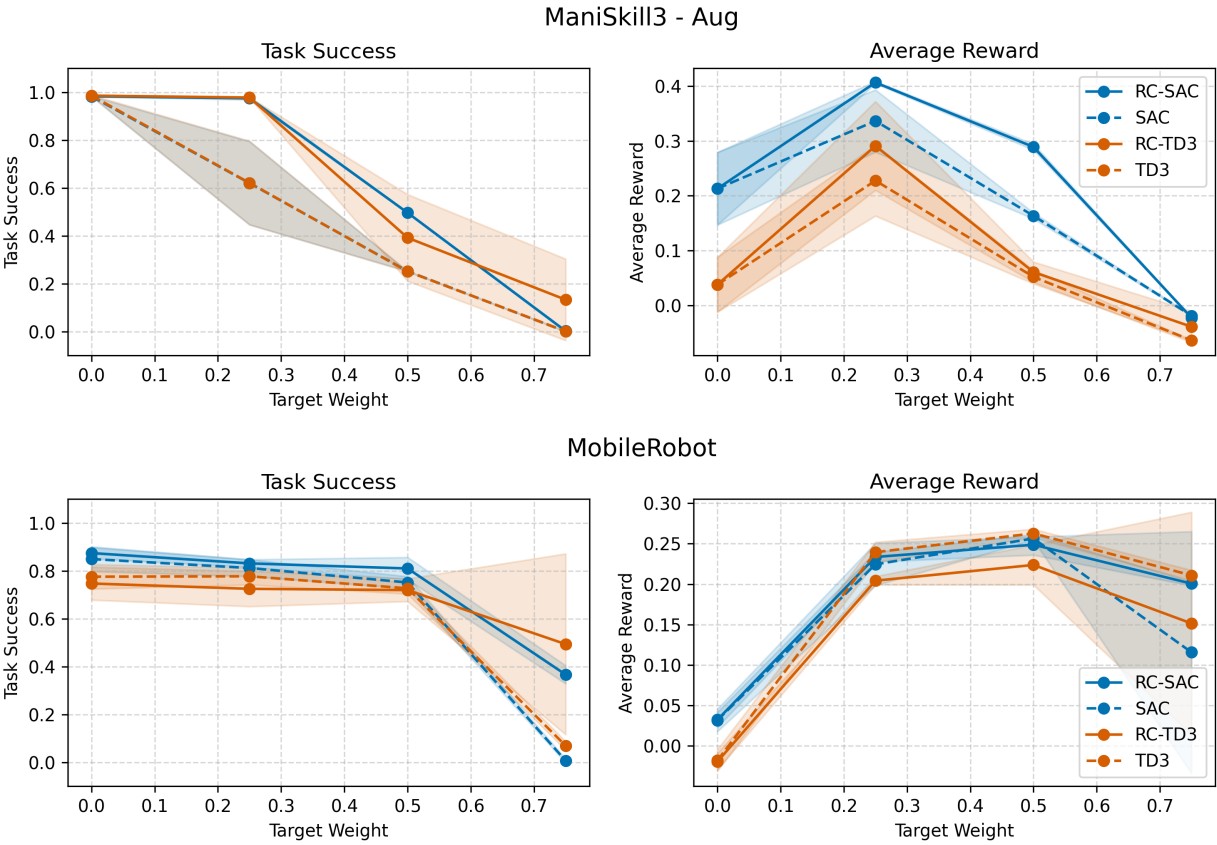

Figure 3: Comparison of the mean success rate and average reward for ManiSkill3 (4 environments aggregated) and MobileRobot with different $w_{\text{target}}$ of RC-TD3 and RC-SAC to their baselines. The average reward is computed using a $w_{\text{target}} = 0.5$ to make results comparable.

## 5.1 How effective is the Two-Stage Curriculum?

The overall results of our reward curriculum framework are presented in Fig. 1. We observe that the curriculum variants (with RC) consistently outperform the baselines trained using a fixed $w_{\text{target}}$ from the beginning. While the average reward often remains similar with occasional improvements, our method substantially increases the task success rate in many cases, thus effectively increasing the priority of learning the task and avoiding reward hacking. Only in a single case (SAC finger turn - easy) the success rate was slightly reduced by the curriculum. Specifically the task success in the augmented DM environments (corresponding to $r_{\text{base}}$) increased from 0.34 to 0.68 and the average $r_{w_{\text{target}}}$ improved from 0.60 to 0.73. Similarly, in MobileRobot the success rate substantially increased from 0.04% to 43% while obtaining a similar average reward. In ManiSkill3-Aug the success rate also substantially increased from 62.1% to 97.6 % and $r_{w_{\text{target}}}$ changed from 0.282 to 0.349. Notably, the results are obtained without environment-specific tuning of curriculum hyperparameters, as the same values are used for all experiments. Our method becomes especially effective in environments such as finger spin or PullCubeTool where the auxiliary terms seem to drastically hinder exploration leading to baselines not learning the task at all. Instead, the curriculum versions obtain near-perfect performance, vastly outperforming the baseline. For detailed results per environment, we refer to Table 1 in the Appendix.

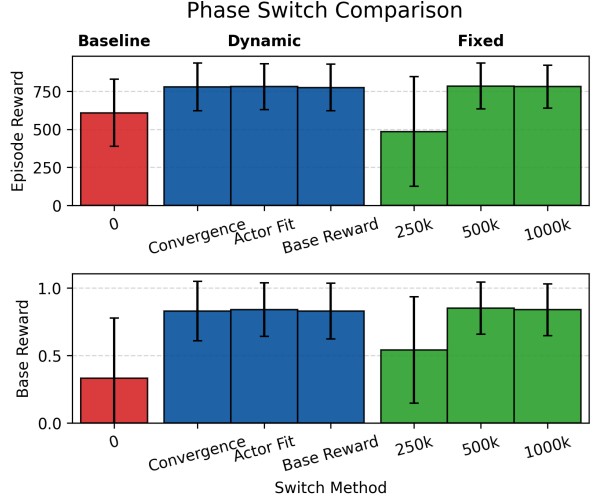 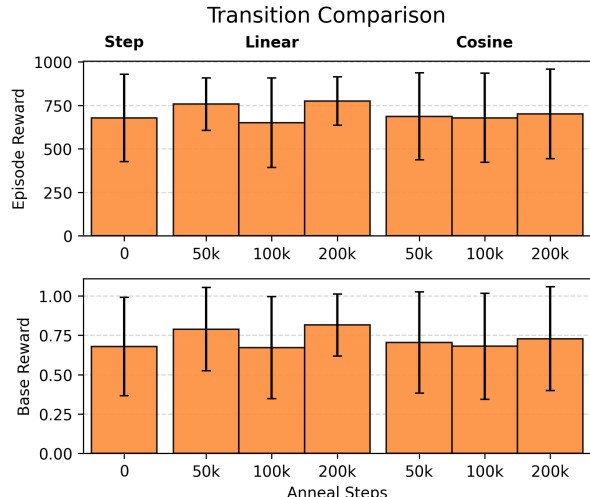

(a) Comparison of the mean episode reward of RC-SAC with different switches as described in Section 4.4. The values are average over the last 50 [k] training steps.

(b) Comparison of the mean episode reward for different transitions as described in Section 4.5. The values are average over the last 50 [k] training steps.

## 5.2 How robust is the method to different $w_{\text{target}}$?

One important aspect of the reward curriculum method is its robustness to different $w_{\text{target}}$. In order to test this, we run experiments for the ManiSkill3 environments with $w_{\text{target}} \in [0.0, 0.25, 0.5, 0.75]$ with 3 random seeds. The results are displayed in Fig. 3. Both in terms of Task success and average reward, the curriculum method clearly outperforms baseline SAC and TD3. Interestingly, already a small target weight of 0.25 substantially lowers the performance of both SAC and TD3 while RC-SAC and RC-TD3 manage to remain 100% success. For higher weights also the curriculum versions obtain reduced success rates, while still being clearly better to the baselines until both end up with near 0% success rates for $w_{\text{target}} = 0.75$.

We repeat the same experiment for the Mobile Robot environment with $w_{\text{target}} \in [0.0, 0.25, 0.5, 0.75]$ for 3 random seeds. The results are shown in the bottom half of Fig. 3. Also in this case the success rate of the curriculum versions is higher or better compared to their baselines. Interestingly RC-TD3 consistently obtains a slightly lower average reward compared to TD3 while being equal or better in terms of success. This is largely due to TD3 taking smaller actions and preferring slightly longer episodes compared to RC-TD3.

## 5.3 Ablation Studies

To better understand the impact of our method's components, we perform the following ablation studies. They are conducted on a subset of the environments that should capture the main trends, namely on finger spin, walker run, reacher easy, and on PullCubeTool-v1.

### 5.3.1 When should the second phase be initiated?

Firstly, we want to analyze how sensitive our method is to the precise choice of when to change curriculum phases. For that, we compare the different methods presented in Section 4.4 with fixed switches at 250 [k], 500 [k], and 1000 [k] steps. During this comparison, we will fix the transition to linear annealing $w$ towards $w_{\text{target}} = 0.5$ over 100 [k] steps and choose RC-SAC only. We expect similar results for RC-TD3. Results averaged over the last 50 [k] steps of training are shown in Fig. 4a. As it can be seen, all switches reach similar performance except the fixed switch at 250 [k] steps. However, even that switch clearly outperforms the baseline trained using the full reward from the beginning, especially in terms of $r_{\text{base}}$. Therefore, we conclude that the first phase has to be sufficiently long to learn $r_{\text{base}}$, but the precise time to switch after that does not have a significant influence. In continuation, we will use the "Convergence" switch given that its hyperparameters neither depend on the specific algorithm nor environment, making it generally applicable.

### 5.3.2 How critical are the transition dynamics between phases?

Next, we compare different methods to anneal $w$ from 0 to $w_{\text{target}} = 0.5$. As introduced in Section 4.5, we compare instant annealing (0 steps) with linear or cosine annealing over 50 [k], 100 [k], and 200 [k] steps. The aggregated results for RC-SAC and RC-TD3 are shown in Fig. 4b. As it can be seen, the performance for all strategies is largely similar, with a tendency that longer annealing leads to better results. Again, the method seems to be robust to the precise choice, confirming the hypothesis that given that $r_{\text{base}} \subset r_{w_{\text{target}}}$, the transition should be relatively stable in all cases. Given the slightly better results for linear annealing over 200 [k] steps, we used this to obtain the overall results presented in Fig. 1.

### 5.3.3 What is the importance of a flexible replay buffer?

Further, we compare resetting the network weights or resetting the replay buffer (i.e. deleting all samples collected so far) after switching curriculum phases to our method, which reuses past experience via the flexible replay buffer for RC-SAC. For that, we use linear annealing over 200 [k] steps and pick two environments where the method had a substantial impact: finger-spin and PullCubeTool. The results are shown in Fig. 5. As can be seen, resetting the network weights seems to destabilize training right after switching phases. However, after a short while, training becomes stable again and reaches a similar final performance. Similarly, resetting the replay buffer leads to elevated training instabilities, indicated by the increased average reward variance. However, even when resetting the buffer upon switching phases, a substantial amount of samples will be collected with a policy for small $w$ given the slow annealing. This explains the similar performance compared to not resetting. Further, completely removing the flexible replay buffer would require resetting the buffer at every step, thus making the method fully on-policy for the annealing duration. We conclude that both the pretrained networks and the flexible replay buffer increase training stability and are thus a vital part of the method.

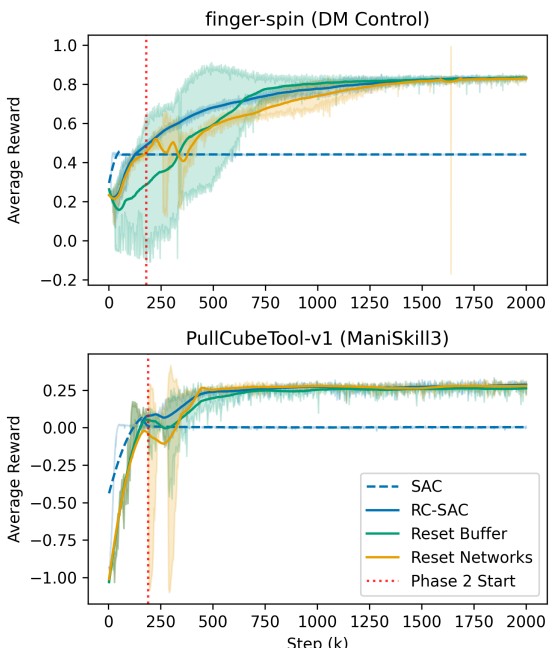

Figure 5: Comparison of RC-SAC with SAC and resetting the buffer or networks when switching between curriculum phases. Results are computed using 2 random seeds and smoothed using a Savitzky-Golay filter with a window length 100 and polyorder 2.

## 6 Conclusion

In this work, we present a two-stage reward curriculum, where we first train an RL agent on an easier task-related subset of rewards and then switch to training using the full reward by also including behavior-related terms. We investigate different methods to automatically decide when to switch from the first to the second phase, how to transition between phases, and the importance of a flexible replay buffer that adaptively calculates the current rewards to reuse samples between phases. In extensive experiments, we show that our method outperforms the baseline trained on the full reward from beginning in almost all cases, especially in terms of task success. Further, we find that the reward curriculum is most effective for environments where the behavioral objectives substantially hinder exploration as it avoids reward hacking. We believe that this work contributes towards developing stable RL methods to effectively learn challenging objectives as, for instance, in robotics, where multiple conflicting objectives are often inevitable.

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

## A    Environments

### A.1    DM Control Suite

The DM Control Suite (Tassa et al., 2018) is a well-known collection of control environments in RL. In our experiments we include the following ones: walker-run, reacher-easy, finger-spin, walker-walk, fish-upright, reacher-hard, finger-turn-hard, finger-turn-easy, ball-in-cup-catch, and cheetah-run. We use the state-space observations and the environments have an action space $\mathcal{A} \in [-1, 1]^2$. The original rewards are in the range $[0, 1]$, which we take as base reward $r_{\text{base}}$. Furthermore, we introduce the following reward term to minimize action magnitudes, to find efficient solutions to the control problems

$$r_{\text{aux}} = -\sum_{i=1}^{d} |a_i| \tag{8}$$

where $d$ is the number of actions of each environment. This is used as an auxiliary term in the curriculum reward (see Equation 1).

### A.2    ManiSkill3

To encourage safe and smooth robot behavior suitable for real-world deployment, we extended the reward function in ManiSkill3 (Tao et al., 2025). Additional to the task-specific terms we introduce behavioral regularization terms. The behavioral terms penalize undesirable motion characteristics and are defined as follows:

**Smoothness:**   Penalizes abrupt changes in action commands to encourage smooth control signals:

$$r_{\text{smooth}} = -\|\mathbf{a}_t - \mathbf{a}_{t-1}\|_2 \tag{9}$$

where $\mathbf{a}_t$ is the action at timestep $t$.

**Jerk:**   Penalizes changes in joint velocity to reduce mechanical stress and vibration:

$$r_{\text{jerk}} = -\|\dot{\mathbf{q}}_t - \dot{\mathbf{q}}_{t-1}\|_2 \tag{10}$$

where $\dot{\mathbf{q}}_t$ denotes the joint velocities at timestep $t$, excluding gripper joints.

**Effort:**   Penalizes large action magnitudes to encourage energy-efficient motions:

$$r_{\text{effort}} = -\|\mathbf{a}_t\|_2 \tag{11}$$

The base reward is the original reward and the auxiliary reward is defined as the sum of the behavioral terms as

$$r_{\text{aux}} = r_{\text{smooth}} + r_{\text{jerk}} + r_{\text{effort}} \tag{12}$$

### A.3    Mobile Robot

To evaluate our method in a more realistic and complex setting (in terms of rewards), we consider a mobile robot navigation problem where the robot is tasked to reach a goal position while avoiding obstacles and satisfying various other objectives. The environment maps are randomized and contain both permanent (e.g.,

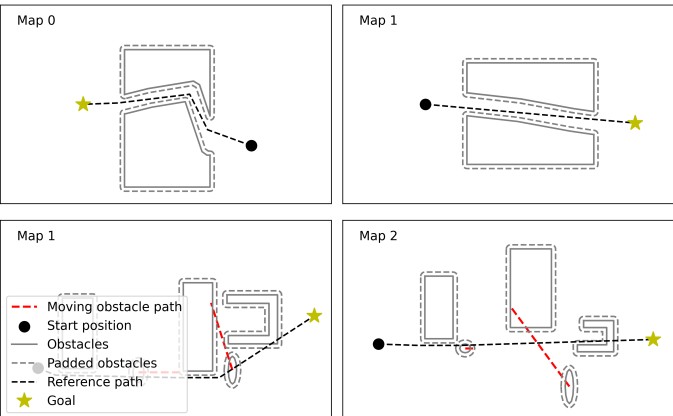

Figure 6: Exemplary environment maps used for training. Obstacle positions, paths, initial states, and goal positions are randomized. While maps 0 and 2 contain dynamic obstacles, maps 1 and 3 only contain static ones.

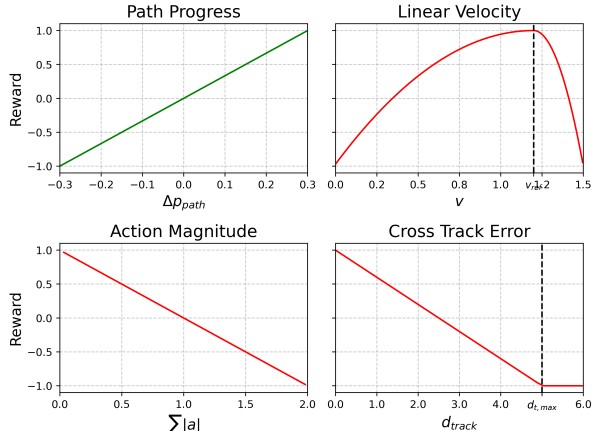

Figure 7: Functions for dense reward terms with $\kappa = 0.942$, $v_{\text{ref}} = 1.2$ and $d_{\text{track,max}} = 5$. The range for each term is normalized to $[-1, 1]$. Green shows the reward shaping term that enables finding the goal faster. The other penalizing terms are colored in red.

walls and corridors) and temporary (e.g., dynamic or static) obstacles. A subset of exemplary environment maps can be found in Fig. 6. Furthermore, a reference path that considers only permanent obstacles is computed using A* (Hart et al., 1968). This reference path should simulate a setting where an optimal path is pre-computed but the robot might not be able to naively follow it due to temporary obstacles.

The state space $\mathcal{S} \in \mathbb{R}^{178}$ consists of the latest two lidar observations, the robot's position, the current speed, the reference path, and the goal position. The action space $\mathcal{A} \in [-1, 1]^2$ represents the translational and angular acceleration.

For the reward design, we will focus on easily interpretable formulations. The objectives include reaching a goal position while driving at a reference velocity, staying close to the reference path, and creating smooth trajectories. Thus, there are three possible outcomes of an episode: (1) reach the goal, (2) timeout, i.e. reach maximum steps, or (3) collide with an obstacle. Empirical tests have shown that penalizing collisions mainly hinders exploration and does not lead to better final policies. Therefore, the only outcome-based reward

included is:

$$r_g = \begin{cases} 10 & \text{if reached goal} \\ 0 & \text{otherwise} \end{cases}$$

The other objectives can be expressed as dense terms evaluated at each step. To enable intuitive weighting, we normalize each term between $[-1, 1]$. We chose this range over $[-1, 0]$ to discourage the policy from learning to crash immediately.

To achieve smooth trajectories we encourage minimizing accelerations. As this corresponds to our action space, we penalize high action values, similar to the DM control experiments, as

$$r_a = 1 - \sum_{i=1}^{2} |a_i|, \quad r_a \in [-1, 1]$$

Further, to drive at a desired reference velocity $v_{\text{ref}}$ we model a velocity reward as a piece-wise quadratic function centered on $v_{\text{ref}}$, and scaled to our desired range. This choice is motivated by the intuition that slowing down is less severe than going too fast.

$$r_v = 1 - \frac{l_2^{\kappa}(v_t - v_{ref}) \cdot 2}{max(l_2^{\kappa}(-v_{ref}), l_2^{\kappa}(v_{max} - v_{ref}))}, \quad r_v \in [-1, 1]$$

with the piece-wise quadratic function $l_2^{\kappa}$ being defined as:

$$l_2^{\kappa}(x) = \begin{cases} \kappa x^2 & \text{if } x > 0 \\ (1 - \kappa)x^2 & \text{otherwise} \end{cases}$$

where $\kappa \in [0, 1]$ controls the slope of the negative and positive regions and $v_{max} = 1.5$ is the maximum velocity.

Further, we linearly penalize deviating from the reference trajectory with the following reward term

$$r_x = \text{clip}\left(\frac{|d_{track}|}{d_{track,\max}}, -1, 1\right), \quad r_x \in [-1, 1]$$

where $d_{track}$ is the distance between the agent and the reference trajectory and $d_{track,\max}$ is a tuning parameter that sets the maximum distance. This is done to make the robot's behavior predictable, such that it only deviates from a planned path if necessary.

To enable effective learning we additionally make use of potential-based reward shaping (Ng et al., 1999), by encouraging progress along the reference path, through

$$r_p = \frac{p_{\text{path}}(s') - p_{\text{path}}(s)}{v_{\max} \cdot dt}, \quad r_p \in [-1, 1]$$

where $p_{\text{path}}(s)$ is the position on a path in state $s$ and the denominator is the maximum distance traversed in one step. As it is potential-based, it does not alter the ordering over policies (Ng et al., 1999). Fig. 7 gives an overview of the dense reward terms used.

## A.4 Reward formulation

As the outcome reward is very sparse and keeps important information we exclude it from the curriculum. Thus the overall reward is given by

$$r_w = r_g + (1 - w) \cdot r_{\text{base}} + w \cdot r_{\text{aux}} \tag{13}$$

where

$$r_{\text{base}} = 0.02 \cdot r_p \tag{14}$$

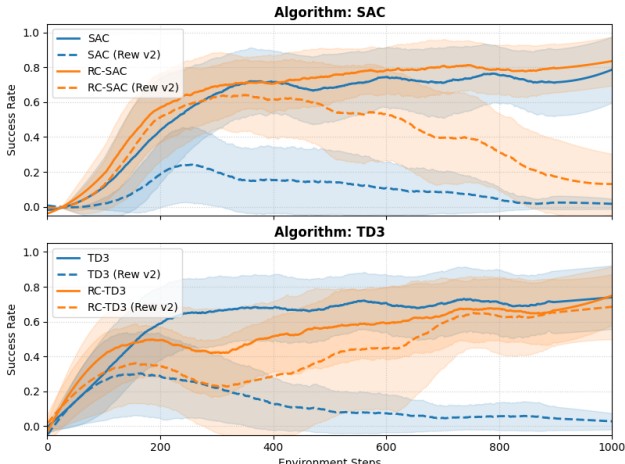

Figure 8: Different reward formulations for MobileRobot for 3 random seeds. Results are smoothed using a Savitzky-Golay filter with window length 200 and polyorder 2.

and

$$r_{\text{aux}} = 0.5 \cdot (r_v + r_a + r_x) \tag{15}$$

are the auxiliary terms.

We compare this to keeping the outcome reward in the curriculum ($r_{\text{base}} = r_g + 0.02 \cdot r_p$) towards the setting above in Fig. 8 for $w_{\text{target}} = 0.5$. As can be seen, the lower outcome reward in phase 2 hinders learning the task for RC-SAC and makes solely optimizing auxiliary terms a better performing policy, thus leading to ill-defined rewards.

## B  Additional Results

### B.1  Convergence Switch Parameters

We ablate different parameters for the proposed convergence-based switch described in Section 4.4 for 1000k training steps for finger-spin and walker-run. As can be seen in Fig. 9, the exact choice of horizon $h$ and slope threshold $\epsilon_{\text{slope}}$ does not substantially alter the results. Thus the method is robust to different hyperparameter choices and in line with other experiments, that suggest that the exact time of switching is not vital as long as it is not too early.

### B.2  Numeric results

The average rewards obtained using SAC, RC-SAC, TD3, and RC-TD3 presented in Fig. 1 are presented in detail in Table 1.

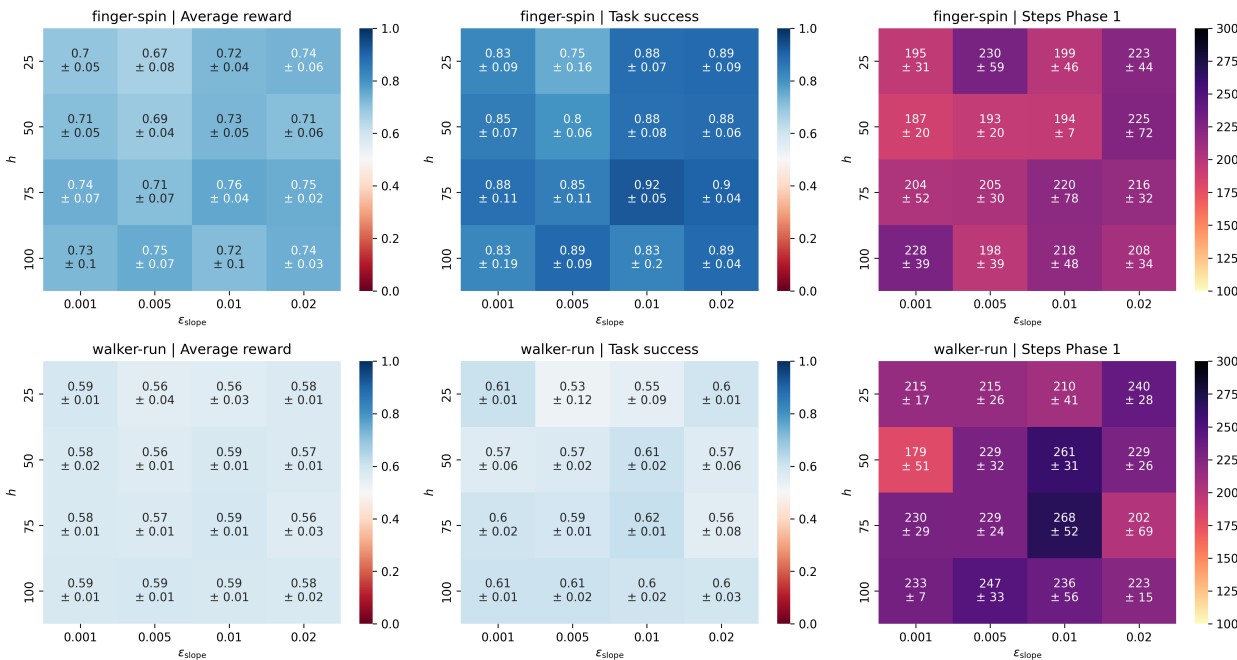

Figure 9: Different values of slope horizon $h$ and threshold $\epsilon_{\text{slope}}$ tested on finger-spin and walker-run environments.

| Method
Environment | TD3 | RC-TD3 | SAC | RC-SAC |
|---|---|---|---|---|
| finger-turn-easy | $0.56 \pm 0.05$ | $0.49 \pm 0.02$ | $0.71 \pm 0.04$ | $0.65 \pm 0.05$ |
| reacher-hard | $0.76 \pm 0.13$ | $0.82 \pm 0.15$ | $0.90 \pm 0.01$ | $0.91 \pm 0.01$ |
| reacher-easy | $0.91 \pm 0.02$ | $0.92 \pm 0.01$ | $0.92 \pm 0.01$ | $0.91 \pm 0.02$ |
| fish-upright | $0.76 \pm 0.03$ | $0.79 \pm 0.07$ | $0.77 \pm 0.03$ | $0.77 \pm 0.06$ |
| finger-turn-hard | $0.37 \pm 0.20$ | $0.35 \pm 0.25$ | $0.54 \pm 0.09$ | $0.56 \pm 0.04$ |
| walker-run | $0.47 \pm 0.00$ | $0.62 \pm 0.00$ | $0.45 \pm 0.00$ | $0.57 \pm 0.04$ |
| walker-walk | $0.85 \pm 0.01$ | $0.86 \pm 0.01$ | $0.81 \pm 0.04$ | $0.84 \pm 0.02$ |
| cheetah-run | $0.46 \pm 0.00$ | $0.46 \pm 0.00$ | $0.42 \pm 0.00$ | $0.62 \pm 0.01$ |
| ball-in-cup-catch | $0.83 \pm 0.19$ | $0.94 \pm 0.00$ | $0.92 \pm 0.00$ | $0.92 \pm 0.00$ |
| finger-spin | $0.46 \pm 0.00$ | $0.77 \pm 0.05$ | $0.44 \pm 0.00$ | $0.83 \pm 0.00$ |
| MobileRobot | $0.21 \pm 0.01$ | $0.15 \pm 0.14$ | $0.12 \pm 0.12$ | $0.20 \pm 0.03$ |
| PullCubeToolMO-v1 | $-0.02 \pm 0.00$ | $0.23 \pm 0.00$ | $0.00 \pm 0.00$ | $0.29 \pm 0.01$ |
| PullCubeMO-v1 | $0.26 \pm 0.01$ | $0.21 \pm 0.05$ | $0.41 \pm 0.01$ | $0.37 \pm 0.04$ |
| PickCubeMO-v1 | $0.31 \pm 0.00$ | $0.29 \pm 0.01$ | $0.33 \pm 0.00$ | $0.32 \pm 0.00$ |
| LiftPegUprightMO-v1 | $0.36 \pm 0.27$ | $0.43 \pm 0.26$ | $0.60 \pm 0.22$ | $0.65 \pm 0.06$ |

Table 1: Mean and standard deviation of the average reward for SAC and RC-SAC for all different environments over 3 random seeds average over the last 50 [k] training steps. Here we use $w_{\text{target}} = 0.5$ for DM Control envs, $w_{\text{target}} = 0.25$ for ManiSkill3 and $w_{\text{target}} = 0.75$ for MobileRobot.

