# OpenReview forum: "Decoupling Task and Behavior: A Two-Stage Reward Curriculum in Reinforcement Learning for Robotics"
_TMLR — Under review for TMLR_

### Review · Reviewer_Vccy · 2026-06-21

**Summary Of Contributions:**

This paper proposes a two-stage reward curriculum method in reinforcement learning for optimizing multiple (two) objectives simultaneously. The method first trains the agent on a simplified task-only reward function and then introduces the full reward that includes auxiliary behavior-related terms. Experiments are conducted on DeepMind Control Suite, ManiSkill3, and a mobile robot environment.

**Audience:**

No

**Audience Explanation:**

The proposed reward curriculum and replay-based reward recomputation are simple and potentially useful ideas for practitioners. However, the paper considers a relatively restricted setting in which the task and auxiliary objectives are clearly defined and the target weight $w_{\text{target}}$ is fixed and given. More importantly, as noted above, the paper does not compare its method with existing reward-curriculum approaches. Because the paper frames its contribution as a strong reward-curriculum method without sufficiently supporting this claim, its current interest and impact for the broader TMLR audience are limited.

**Broader Impact Concerns:**

None.

**Claims And Evidence:**

No

**Claims Explanation:**

The paper clearly explains the proposed method and provides a concise intuition for its design. The empirical evaluation offers reasonably clear evidence that the proposed reward curriculum can improve over directly optimizing the fixed target reward from the beginning, across both SAC and TD3. The ablations on phase switching, annealing, and replay-buffer handling are also useful.

However, the paper is framed as proposing a reward-curriculum method with superior performance over relevant baselines, yet the evidence is insufficient to support such a broad claim. In particular, the paper does not compare against prior reward-curriculum methods or even strong simple alternatives, such as a fixed linear ramp of the auxiliary-reward weight from the beginning of training. Several relevant reward-curriculum works are discussed in the introduction and related-work section, but none are included in the experimental evaluation. The paper also does not provide a sufficiently convincing justification for why such comparisons are unnecessary. In particular, the paper notes that Pathare et al. (2025) study a three-stage curriculum and find reward curricula largely ineffective. This observation makes a direct comparison or a more careful analysis especially important. If prior reward curricula failed while the proposed method succeeds, understanding the reason would be scientifically valuable. For example, is the difference due to the task/behavior reward decomposition, the switching and annealing strategy, replay-based reward recomputation, or properties of the modified benchmark rewards? The current experiments do not explore these possibilities.

**Requested Changes:**

The paper should avoid claiming general superiority over reward-curriculum methods unless it evaluates against representative alternatives. If the proposed approach does not outperform stronger reward-curriculum baselines, then it would need to change its contribution framing. It should instead be reframed as something like a simple, practical, and easy-to-use reward-curriculum method for robotic tasks in which auxiliary behavioral penalties suppress exploration.

To support this reframing, in addition to characterizing its behavior and comparing it to baselines, the paper should also provide practical guidance for when it should be used and when it fails. Relevant evaluation dimensions may include task success rate, final scalarized reward, sample efficiency, stability across random seeds, hyperparameter sensitivity, computational complexity, and failure modes. A method that is slightly weaker than a more sophisticated baseline may still provide a meaningful contribution if it is substantially simpler, more stable, more sample-efficient, or less sensitive to hyperparameters.

Finally, the paper would be considerably stronger if it systematically studied the relationship between the base task and the auxiliary reward term. In particular, it should clarify when the proposed curriculum helps, when it is unnecessary, and when it fails, for example, when the auxiliary term strongly suppresses exploration, when the base-task reward alone is insufficient for learning, or when the base-task and final objectives require qualitatively incompatible behaviors.

---

### Review · Reviewer_QWed · 2026-06-22

**Summary Of Contributions:**

## Summary

This paper proposes a two-stage reward curriculum for reinforcement learning in robotics when the desired reward combines task-completion terms with auxiliary behavioral objectives, such as energy efficiency, smoothness, or low jerk. The main idea is to first train the agent using only the base/task reward so that it can discover task-solving behavior, and then gradually anneal toward the full reward that includes auxiliary terms. The method is instantiated for TD3 and SAC, with a replay buffer that stores the base and auxiliary reward components separately so that prior samples can be reused under the current curriculum weight. The paper evaluates the approach on DM Control, ManiSkill3, and a mobile robot navigation environment, and reports improved performance and robustness relative to training directly on the full reward.

## Strengths

1. The proposed algorithm is clearly stated and easy to understand. The two-stage formulation is simple, practical, and well motivated for robotics settings where auxiliary penalties can discourage exploration before the task is learned.

2. The paper provides a broad empirical evaluation across several environments and two standard off-policy algorithms, TD3 and SAC. The experiments help illustrate when the reward curriculum improves over directly optimizing the full reward.

3. The ablation studies on switching criteria, annealing schedules, and replay-buffer reuse are useful. They clarify which design choices matter most and suggest that reusing samples across phases is an important part of the method.

## Weaknesses

1. The paper provides useful intuition and a target-shift bound, but it does not provide a formal convergence guarantee for the proposed curriculum algorithm. It would be helpful to more clearly state what is theoretically justified versus what is supported empirically.

2. Because the method requires a first stage before optimizing the full reward, the total training process may be slower in settings where the full reward is already learnable. The paper would benefit from a clearer discussion of the computational cost and sample-efficiency trade-off introduced by delaying the second stage.

3. The novelty relative to standard TD3 and SAC could be made more explicit. From the algorithmic description, the main changes appear to be the curriculum-dependent construction of $r_w$ and the storage/reuse of separate reward components in the replay buffer. Highlighting these modifications more directly would make the contribution easier to assess.

**Audience:**

Yes

**Audience Explanation:**

The research topic on RL for robotics is meaningful.

**Claims And Evidence:**

Yes

**Claims Explanation:**

The claims are supported by simulation results.

**Requested Changes:**

1. What is the relationship between the original reward $r$ in the MDP definition and the reward $r_w$ defined in Eq. (1)? Is $r_w$ intended to be the actual reward optimized by the algorithm, or is it a curriculum-specific surrogate for a fixed target reward?

2. In Line 6 of Algorithm 1, what exactly is meant by "env steps"? Relatedly, in Line 8, what does "Step env" denote? I assume "env" stands for environment, but the distinction between environment steps, gradient steps, and ordinary time steps could be explained more clearly.

3. In Line 30 of Algorithm 1, what are the "target parameters," and how are they updated?

4. In Section 4.1, please highlight which parts are new relative to the original TD3 and SAC algorithms. Is the only change replacing $r$ with $r_w$ in the Bellman target, or are there additional changes besides reward-component storage and curriculum scheduling?

5. How should practitioners decide whether the extra first-stage training cost is worthwhile for a new environment? Are there diagnostics that indicate when direct full-reward training is likely to fail due to the auxiliary terms?

---

### Review · Reviewer_dHta · 2026-07-15

**Summary Of Contributions:**

The manuscript introduces a two-stage reward curriculum framework for off-policy reinforcement learning. In the proposed setting, the agent first learn to optimise a purely task related reward, and subsequently anneals towards a target reward function that additionally considers behavioural objectives (such as constraints on the jerk/acceleration or the total energy spent during task execution).
The authors evaluate several strategies for transitioning between the two curriculum stages (three adaptive criteria and several fixed-time strategies), and the method is implemented both in in Soft Actor-Critic and Twin Delayed Deep Deterministic Policy Gradient.

The main idea is, instead of abrupt switching to the more complex reward function, to slowly anneal the additional term when a performance criterion has been satisfied for the first stage.

The authors demonstrate the effectiveness of their method on evaluated on twelve DeepMind Control tasks, four augmented ManiSkill3 manipulation tasks, and a mobile-robot navigation environment.

The main insight from their results is that the proposed curriculum quite often substantially improves task completion, when considering auxiliary penalties would otherwise suppress the exploration required to discover the parameter sets that support the intended behaviour.

Overall, the paper tackles an interesting and rather important problem, however I am a bit hesitant about the contribution of the submission, since as I understand this is the proposal of annealing the auxiliary reward term. Moreover there are also some mathematical errors/ imprecisions that I detail below.


## Strengths

- The problem the authors tackle is interesting and relevant for multiple RL applications.

- The solution they propose is rather simple and easy to implement in multiple settings.

- The authors test several switch and annealing choices.

## Weaknesses

- The claim regarding the novelty of the two stage training is a bit exaggerated. The manuscript itself cites prior work that progressively introduces or increases reward components, also see reference [1] below.

- Three random seeds are insufficient for the strength of the claims

- The term "task manifold" is not properly defined.

-  The main baseline the authors present is slightly insufficient.

- Eq. 4 seems to have a wrong sign in front of the entropy term, which would result in entropy reduction.

- The mobile-robot path-deviation reward seems to have opposite sign. The $r_x$ seems to take values form 0 to 1 (due to the absolute in the nominator) and we would want to penalise large deviations, thus a minus in front is needed.

- The potential-based shaping term does not include the discounting factor $\gamma$.
----

**Audience:**

Yes

**Audience Explanation:**

Yes. The paper tackles an important problem in reinforcement learning for robotics, that of learning effectively from complex reward functions that combine task and behavioural objectives. The proposed curriculum is simple, easily applicable to general off-policy methods, and the empirical results suggest practical potential. Despite my concerns about the strength of the evidence and some theoretical claims, I believe the work would be of interest to researchers working on reinforcement learning, robotics, and curriculum learning.

**Claims And Evidence:**

No

**Claims Explanation:**

The paper provides some empirical evidence that initiating training with the task loss function followed by the full reward function improves task acquisition relative to training with the full loss from the beginning.

However, sever claims are not supported by convincing and clear evidence. I outline them below:

- The authors claim that their work is the first study that systematically investigates a two-stage curriculum. However given the paper [1] mentioned below this is not correct.
- The main baseline the authors present is slightly insufficient. The main baseline trains SAC or TD3 on the final reward function from the beginning of training. This demonstrates that the proposed schedule can outperform one difficult optimization strategy, but it does not isolate why it works and does not necessarily establish outperformance of plausible alternative approaches.

-  The claim that replay-buffer reuse is “critical” is not really demonstrated by the ablation: resetting the buffer produces similar final performance, and the comparison uses only two seeds.

- Most experiments use only three seeds, which is insufficient for strong conclusions in high-variance reinforcement-learning settings.
- The paper demonstrates improved task success, but does not consistently show that the resulting successful policies better optimize the intended behavioural quantities such as effort, acceleration, or jerk.

**Requested Changes:**

r


- The authors should cite the following papers and discuss how their proposed framework of two stage reward curriculum compares to the two stage curriculum introduced in [1]. Moreover the authors should moderate their claim in  the final sentence of Section 2 Related work, where they claim to be the first work that systematically investigates a two-stage curriculum. In light of this paper, they should refine the statements about the contributions of the present paper and they should not imply that separating task acquisition from behavioural optimization is fundamentally unprecedented. A stronger related-work comparison is required, including direct empirical baselines implementing the most relevant previously proposed reward curricula already cited in the submission.

- Apart from competing baselines, I would find helpful to disambiguate why the method actually works to compare also with the following alternatives:
  - Augment reward function without annealing: Train on $r_{\mathrm{base}}$, then instantaneously switch to the final reward while retaining the networks and buffer.
  - Fixed-time two-stage curriculum: The adaptive convergence detector adds complexity, but fixed switches at 500\,000 and 1\,000\,000 steps appear to perform similarly.
  - Linear annealing from the start of training
  - Reward normalization or adaptive scaling: The initial difficulty to train may arise partly from incompatible reward magnitudes rather than conflicting objectives. Can you also show or verify that gradients inital at the training are conflicting while later on when the base reward term is sufficiently optimised they are more aligned?

- How do you guarantee preservation of task performance at second stage? Did you test for it? Couldn't it be that at second stage the reward from the base term decreases but total reward still increases due to the second term? Can you report objective components separately for each setting? I.e. for each setting and environment show P(success), $J_{base}$, $J_{aux}$, $J_{aux} | \text{success}$

- The central mechanism for the approach according to the authors claims seems to be  improved exploration. This should be somehow quantified.

-   Is there any  guarantee for convergence to the optimum of the final reward?

- Does the curriculum genuinely produce smoother, lower-effort, or more energy-efficient successful trajectories, or does it merely preserve task performance despite the auxiliary penalty?

- Increase the number of seeds to 5 or 10.

----
### References

[1] Freitag, K., Ceder, K., Laezza, R., Åkesson, K., & Chehreghani, M. H. (2024). Curriculum Reinforcement Learning for Complex Reward Functions. arXiv preprint arXiv:2410.16790.